# Prismatic Synthesis:
# Gradient-based Data Diversification
# Boosts Generalization in LLM Reasoning

**Jaehun Jung**[1,2]   **Seungju Han**[1*]   **Ximing Lu**[1,2*]   **Skyler Hallinan**[3*]
**David Acuna**[1]   **Shrimai Prabhumoye**[1]   **Mostofa Patwary**[1]
**Mohammad Shoeybi**[1]   **Bryan Catanzaro**[1]   **Yejin Choi**[1]
[1]NVIDIA Research
[2]University of Washington
[3]University of Southern California
`hoony123@cs.washington.edu`[*]

## Abstract

Effective generalization in language models depends critically on the diversity of their training data. Yet existing diversity metrics often fall short of this goal, relying on surface-level heuristics that are decoupled from model behavior. This motivates us to ask: *What kind of diversity in training data actually drives generalization in language models—and how can we measure and amplify it?* Through large-scale empirical analyses spanning over 300 training runs, carefully controlled for data scale and quality, we show that data diversity can be a strong predictor of generalization in LLM reasoning—as measured by average model performance on unseen out-of-distribution benchmarks. We introduce **G-Vendi**, a metric that quantifies diversity via the entropy of model-induced gradients. Despite using a small off-the-shelf proxy model for gradients, G-Vendi consistently outperforms alternative measures, achieving strong correlation (Spearman's $\rho \approx 0.9$) with out-of-distribution (OOD) performance on both natural language inference (NLI) and math reasoning tasks. Building on this insight, we present **Prismatic Synthesis**, a framework for generating diverse synthetic data by targeting underrepresented regions in gradient space. Experimental results show that Prismatic Synthesis consistently improves model performance as we scale synthetic data—not just on in-distribution test but across unseen, out-of-distribution benchmarks—significantly outperforming state-of-the-art models that rely on 20 times larger data generator than ours. For example, PrismMath-7B, our model distilled from a 32B LLM, outperforms R1-Distill-Qwen-7B—the same base model trained on proprietary data generated by 671B R1—on 6 out of 7 challenging benchmarks.

## 1 Introduction

The idea that diverse training data leads to better language models is both intuitive and widely acknowledged—a growing body of works support this intuition, linking data diversity to improved robustness, sample efficiency, and generalization [2, 45, 69]. But despite this growing consensus, a fundamental question remains surprisingly underexplored: what kind of diversity actually matters for generalization, and how should we empirically quantify it? While the importance of diversity is broadly recognized, its precise quantification remains elusive—existing measures often rely on task-specific heuristics, or intrinsic textual features such as semantic or lexical variations [58, 67, 18, 19].

---

[*]SH, XL and SH are co-second authors. Project Page: Link

39th Conference on Neural Information Processing Systems (NeurIPS 2025).

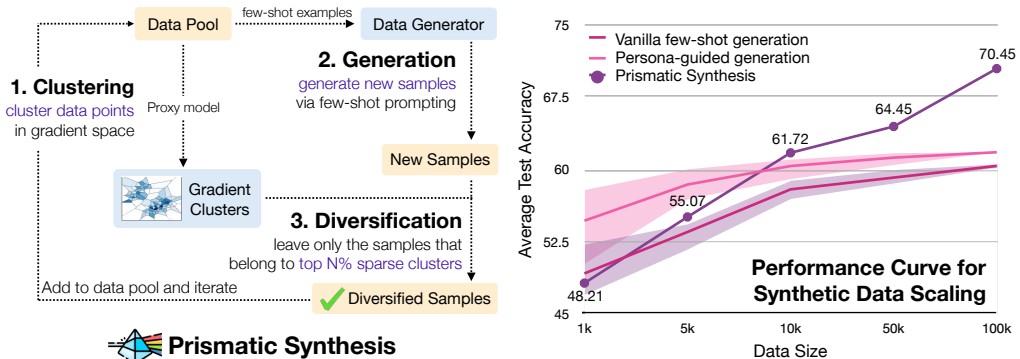

Figure 1: (Left) **Overview of Prismatic Synthesis**. We iteratively (1) cluster samples in a gradient space, (2) generate new samples, and (3) add to the pool only the samples in sparse clusters, consistently improving both the diversity and scale of generated dataset. (Right) Naive scaling of synthetic math data—with no diversification or with a heuristic persona-guided prompting [9]—faces early saturation, when measuring average performance across 7 distinct benchmarks. **Prismatic Synthesis consistently improves model performance beyond 100K, up to million scale synthetic data.** See §3.2.2 for more details.

Yet it remains unclear to what extent these measures actually correlate with downstream model performance—particularly for reasoning tasks, where surface-form diversity offers limited insight.

In this work, we aim to investigate the role of data diversity in LLM reasoning, and how to measure and promote diversity during data curation. To isolate the effect of diversity, we conduct a comprehensive suite of experiments that control for data scale and quality, allowing us to systematically examine how diversity measures relate to model empirical generalization. Our study focuses on post-training scenarios in reasoning tasks, and builds on an extensive empirical framework—generating a synthetic data pool of over 3 million samples and fine-tuning more than 300 models on distinct datasets with varied range of diversity and scale. In both discriminative NLI and generative math reasoning tasks, our analyses reveal three key insights:

- Unless tailored to a specific task, **existing diversity measures often only show moderate to weak correlation with how the model performs on unseen benchmarks**—as they latch onto surface-form features that may not be relevant to the target task.

- We propose G-Vendi, a novel measure that computes entropy of a given dataset in a gradient space. Despite using a small, off-the-shelf proxy model to compute gradients, **G-Vendi strongly correlates (Spearman's $\rho \approx 0.9$) with how the resulting model performs on OOD benchmarks.**

- Diversity often outweighs the scale for out-of-distribution generalization—training on a small dataset with higher diversity can outperform datasets of 10 times larger scale, even when the datasets are drawn from the same data pool. However, **scale is a dominant factor for in-distribution performance, improving models in complementary ways to diversity.**

Leveraging these insights, we seek to further enhance data diversity by strategically generating synthetic data. We find that naive scaling of synthetic data does not improve the model, facing early saturation even with heuristic diversification such as persona-guided prompting [9]. We instead propose **Prismatic Synthesis**, a novel algorithm to scale synthetic data while simultaneously improving the diversity of generated samples. By iteratively (1) clustering existing data in gradient space and (2) rejection-sampling new data that correspond to the sparse clusters, Prismatic Synthesis constantly improves both the diversity and scale of the resulting dataset. Consequently, this simple process empowers consistent improvement in model reasoning beyond million-scale synthetic samples.

We apply our method using off-the-shelf 32B / 72B LLMs to generate two synthetic datasets: PrismMath and PrismNLI. While our datasets are **entirely model-generated without any human annotations**, they produce surprisingly strong models—**often surpassing state-of-the-art models distilled from a 671B teacher-generated trajectories, further verified by ground-truth answers.** For example, our math reasoning model PrismMath-7B yields state-of-the-art results on 6 out of 7 benchmarks (*e.g.,* 57.08% on AIME24 with only SFT), outperforming R1-Distill-Qwen-7B trained on

proprietary data generated by 671B R1 [5]. Overall, these results suggest that strategic diversification may offer greater gains than costly, manual data curation, and highlight the importance of data diversity as a key driver of model generalization.

## 2 G-Vendi: Diversity Measure that Predicts Generalization

Our first goal in this work is to analyze data diversity—often regarded as an intrinsic property of a dataset—through the lens of its extrinsic, practical impact on model performance. In the following sections, we first introduce our novel diversity measure G-Vendi (§2.1), illustrate how we evaluate diversity measures while controlling for confounders (§2.2), and analyze the results (§2.3).

### 2.1  G-Vendi Score

We motivate G-Vendi from the formulation of Pruthi et al. [48] that approximates training data influence with first-order gradients. Let $\nabla l(z; \theta)$ denote the back-propagated loss gradient of a data sample $z$ under model parameters $\theta$. The improvement (*i.e.,* loss reduction) on a test sample $z'$ induced by training on a train sample $z$, is approximately proportional to the dot product between the loss gradients for $z$ and $z'$:

$$l(z'; \theta) - l(z'; \theta') \propto \nabla l(z; \theta) \cdot \nabla l(z'; \theta)$$

In other words, the degree to which training on $z$ helps generalization to $z'$, can be estimated by the similarity between their loss gradients. Extending this insight, we hypothesize and empirically show that when no explicit target distribution of $z'$ is available, promoting the *diversity among the loss gradients of training samples* $z$ may help improve model performance—allowing the training data to cover a broader region of the task distribution, assuming uniform prior over $z'$.

**Collecting Gradient Representation**  G-Vendi implements this idea by utilizing the gradient representation of each sample in a given dataset. Concretely, let $(x, y) \in \mathcal{D}$ denote a training data point where $x$ is the input and $y$ is the output. G-Vendi represents each sample $(x, y)$ with its normalized loss gradient vector computed under an off-the-shelf proxy model $\theta$:

$$g_\theta(x, y) = \frac{-\nabla \log P(y|x; \theta)}{|| - \nabla \log P(y|x; \theta)||} \in \mathbb{R}^{|\theta|} \tag{1}$$

Note here that each $g_\theta(x, y)$ is a $|\theta|$-dimensional vector, which is prohibitively large even with a small proxy model (*e.g.,* 0.5B). We thus follow prior works [61, 57] to reduce the dimension of $g_\theta(x, y)$ from $|\theta|$ to $d << |\theta|$ via Rademacher random projection [46, 50]:

$$g_\theta^{proj}(x, y) = \Pi^T g_\theta(x, y), \quad \text{where } \Pi \in \mathbb{R}^{|\theta| \times d}, \ \Pi_{ij} \sim \mathcal{U}(\{-1, 1\}) \tag{2}$$

The projection qualifies as a Johnson-Lindenstrauss transform [17], and thus preserves the inner product between projected gradients with high probability (while significantly reducing their dimension). We set $d = 1024$ in our experiments, and compute the projection for all samples $(x, y) \in \mathcal{D}$; we denote the collected projections as $G \in \mathbb{R}^{|\mathcal{D}| \times d}$.

Importantly, we set the proxy model $\theta$ to be a small instruction-tuned model (*e.g.,* Qwen2.5-0.5B-Instruct [49]) without any additional training. This greatly simplifies the gradient collection process compared to prior methods, which often require in-distribution warm-up training, weight decomposition, and gradient aggregation across multiple checkpoints [1, 48, 57, 20]. While the exact formulation may be more accurate in estimating influence to a target instance [61], we find that gradients from an off-the-shelf model $\theta$ are still effective in estimating variance *in between* training samples—which does not involve measuring distance to a target distribution. In addition, the efficiency of off-the-shelf gradients makes it suitable for online diversification of large-scale synthetic data, as we discuss in §3.

**Measuring Entropy of Gradients**  We measure the diversity of a dataset $\mathcal{D}$ by computing the entropy of its loss gradients. Specifically, we compute the exponentiated entropy of the normalized covariance matrix of $G$, *i.e.,* the Vendi Score [8] of $G$, allowing us to measure the entropy without knowing the support of the underlying gradient distribution. Let $K$ denote the covariance matrix, *i.e.,* $K_{ij} = \left(GG^T\right)_{ij} / |\mathcal{D}| = g_\theta^{proj}(x_i, y_i) \cdot g_\theta^{proj}(x_j, y_j) / |\mathcal{D}|$. Then the G-Vendi score of $\mathcal{D}$ is computed as:

$$\text{G-Vendi}(\mathcal{D}) = \exp\left(-\sum_i \lambda_i^K \log \lambda_i^K\right) \tag{3}$$

where $\lambda_i^K$ is the $i$-th eigenvalue of $K$. Intuitively, a low value of G-Vendi implies that most of the variance among gradients is explained by a few directional components, indicating low diversity; a high value of G-Vendi implies no dominant directional component among gradients, indicating high diversity. Compared to other aggregation methods (*e.g.,* average pairwise similarity), Vendi score is significantly more compute-efficient when $|\mathcal{D}| >> d$ and robust when features are correlated with each other [8]. G-Vendi is a positive unbounded scalar score, and inherits several desirable properties of the Vendi score for diversity measurement, such as permutation invariance. We further illustrate the properties of our measure in §C.

## 2.2 Evaluating Data Diversity Measures

Our next step is to evaluate how well G-Vendi (and existing diversity measures) can predict model performance on unseen benchmarks. Given a diversity measure $f$ along with two datasets $\mathcal{D}_1$ and $\mathcal{D}_2$ of comparable size and quality, our goal is to evaluate whether $f$ satisfies:

$$f(\mathcal{D}_1) > f(\mathcal{D}_2) \Rightarrow \text{Perf}(\mathcal{M}_1) > \text{Perf}(\mathcal{M}_2) \tag{4}$$

where $\mathcal{M}_1$ and $\mathcal{M}_2$ are the same base models trained on $\mathcal{D}_1$ and $\mathcal{D}_2$, and $\text{Perf}$ estimates the models' task performance on unseen benchmarks. The most straightforward way of evaluating the desideratum is to (1) prepare many datasets with varied levels of diversity, (2) train a model on each dataset, and (3) compute the correlation between performance and diversity. However, translating this evaluation into practice entails non-trivial experimental design, particularly to control for confounding factors such as data quality. Below, we detail our setup focusing on math reasoning and NLI tasks.

**Preparing Datasets**    To control for the effects of data quality, we generate a large pool of synthetic data using LLMs from which we sample distinct training datasets. Compared to off-the-shelf datasets, synthetic data offers several advantages for our experimental setup: it is more scalable, easier to quality-control (by applying the same data-generating pipeline for all samples), and importantly, has been widely adopted for post-training LLMs on reasoning tasks [16, 44, 63].

We take a few-shot generation approach to generate the data pool, by random-sampling 5 demonstrations from a seed dataset then prompting an LLM to generate novel data points. For seed datasets, we use WANLI [28] for NLI, and a mixture of GSM8k [4] and MATH [15] for math reasoning. We prompt Qwen2.5-72B-Instruct [49] to generate both the new problems and corresponding solutions to create a pool of 1.5M samples for each task. Finally, we prepare training datasets by sampling subsets from this data pool. We create 300 distinct subsets with varied levels of diversity, while controlling for their size (N = 100k, 50k, 10k for math reasoning, N = 50k, 10k for NLI). For more details, we refer the readers to §A.1.

**Evaluating Model Generalization**    Yet another challenge lies in defining $\text{Perf}$, *i.e.,* how to estimate the model's empirical performance on unseen benchmarks. One intuitive approach is to prepare multiple unseen benchmarks $B_1, \cdots B_{|\mathcal{B}|}$ and average the accuracies, i.e., $\frac{1}{|\mathcal{B}|} \sum_i Acc_{B_i}(\mathcal{M})$. However, this metric overlooks the fact that not all benchmarks are equally challenging—for example, a 3% improvement on an elementary arithmetic exam should not be deemed equivalent to 3% improvement in an Olympiad-level benchmark, where even the strongest model performs poorly and thus 3% gap is much more significant. To better reflect the benchmark-specific notion of performance gap, we average the relative accuracy of the current model $\mathcal{M}$ against a strong reference model $\mathcal{M}_{\text{ref}}$:

$$\text{Perf}(\mathcal{M}) := \frac{1}{|\mathcal{B}|} \sum_i \frac{\text{Acc}_{\mathcal{B}_i}(\mathcal{M})}{\text{Acc}_{\mathcal{B}_i}(\mathcal{M}_{\text{ref}})} \tag{5}$$

We define $\mathcal{M}_{\text{ref}}$ to be the same base model as $\mathcal{M}$, but trained on the full data pool instead of a subset. Therefore, $\text{Perf}(\mathcal{M}) = 1$ means that the model was able to achieve the same performance as the reference model, despite being trained on a substantially smaller subset.

We initialize $\mathcal{M}$ with Llama-3.2-1B [31] for math reasoning and DeBERTa-v3-large [13] for NLI, and select benchmarks that are considered to be out-of-distribution from our seed datasets. Specifically, we aggregate performance across 7 math benchmarks—SAT-Math, MMLU Elementary, High School,

Table 1: Correlation between model OOD performance and data diversity measures on 100K scale. Compared to baseline measures, **G-Vendi strongly correlates with how the model generalizes to unseen distributions.** The OOD performance for each task is defined as the average relative performance on 7 unseen benchmarks (see §2.2 for more details). For each measure, we fit both log-linear and linear trendlines to the collected data, and report the higher resulting $R^2$.

| Diversity Measure | Math OOD Performance | | NLI OOD Performance | | Math ID Performance | |
|---|---|---|---|---|---|---|
| | $R^2$ | Spearman's $\rho$ | $R^2$ | Spearman's $\rho$ | $R^2$ | Spearman's $\rho$ |
| Embedding Vendi | 0.659 | 0.754 | 0.622 | 0.841 | 0.421 | 0.614 |
| Embedding Dissimilarity | 0.583 | 0.751 | 0.582 | 0.806 | 0.397 | 0.574 |
| 2-gram Entropy | 0.549 | 0.736 | 0.497 | 0.664 | 0.243 | 0.548 |
| Average Perplexity | 0.442 | -0.631 | 0.321 | 0.597 | 0.200 | -0.341 |
| Skill Set Entropy | 0.706 | 0.812 | - | - | 0.527 | 0.695 |
| G-Vendi | **0.823** | **0.899** | **0.791** | **0.893** | **0.697** | **0.780** |

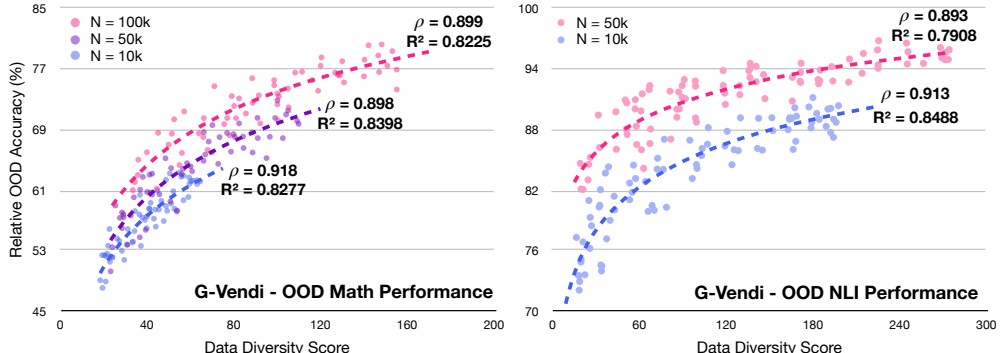

Figure 2: G-Vendi and model OOD performance. **G-Vendi shows a strong log-linear relationship with model performance, when controlling for data scale and quality.** In both tasks, models trained with datasets of high G-Vendi tend to generalize better in OOD benchmarks. Plots for baseline measures are shown in §B.1.

College Math, GSM-IC, Aqua-RAT and Minerva-Math [71, 14, 52, 27, 22]—where the reference model $\mathcal{M}_{\text{ref}}$ achieves meaningful accuracy above 10%—and 7 NLI benchmarks—HANS, WNLI, ANLI, QNLI, NLI Diagnostics, BigBench NLI and ConTRoL [37, 42, 56, 53, 29].

**Baselines** For baselines, we include *Embedding Vendi*: Vendi score with an off-the-shelf embedding model to represent each sample, and *Embedding DisSim*, which measures the average $1 -$ cosine similarity between all pairs of sample embeddings. We use gte-Qwen-7B-Instruct [25], a state-of-the-art embedding model on MTEB benchmark [39]. We also include two traditional metrics *2-gram Entropy* and *Perplexity*, along with an LLM-based metric Skill-Set Entropy for math reasoning—that prompts LLM to extract "reasoning skills" involved in each sample, then measures the entropy of the extracted skill set distributions [6, 33]. We leave further baseline details in §A.1.

## 2.3 Evaluation Results

### 2.3.1 Main Results

**G-Vendi strongly predicts empirical generalization.** Our main results comparing model generalization against diversity measures are shown in Fig. 2 and Table 1. Overall, we find that G-Vendi strongly predicts model generalization—with $R^2 \approx 0.8$ and Spearman's $\rho \approx 0.9$ in both tasks. Notably, the measure outperforms (1) *Embedding Vendi* that utilizes a state-of-the-art embedding model 14 times larger than the proxy model in G-Vendi, and (2) *Skill Set Entropy* which employs GPT-4 and Qwen2.5-72B-Instruct to taxonomize and extract instance-level skill sets. Overall, the result demonstrates that G-Vendi is surprisingly effective in predicting model generalization—due in part to the gradient representation's stronger alignment with task-relevant cognitive processes, as opposed to the surface form features prioritized in the baselines (§2.3.2).

**G-Vendi often outweighs scale for OOD generalization, but scale primarily drives in-distribution performance.** As shown in Fig. 2, training on larger scale datasets generally improves OOD

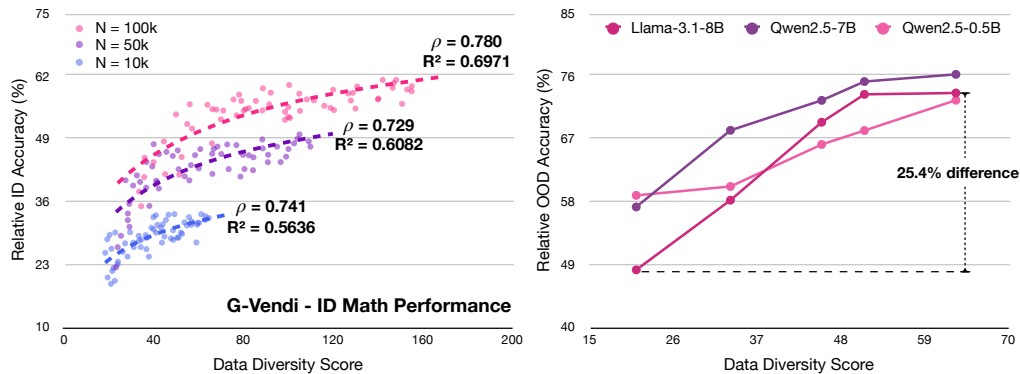

Figure 3: (Left) G-Vendi and in-distribution performance. Compared to OOD, ID performance is more heavily dominated by the scale of the training dataset—*e.g.,* 10K datasets with high diversity are less likely to outperform 50K datasets, compared to OOD results in Fig. 2. But significantly low diversity can still harm in-distribution performance. (Right) Ablation on the student model. **Higher G-Vendi correlates with stronger OOD performance, across model family and scale.**

performance. However, diversity can override the scale, *e.g.,* a 10K dataset with more diversity often outperforms a 100K dataset with less diversity.

We also investigate how data scale and diversity impact in-distribution (ID) performance. In Fig. 3, we evaluate the same model checkpoints as in Fig. 2, but on the in-distribution test set of MATH and GSM8k. Interestingly, while data diversity does correlate positively with ID performance, the strength of correlation is consistently weaker than on OOD across all data scale. While significantly low diversity (*e.g.,* below 40 for $N = 100k$) does harm model's ID performance, above this level, higher data diversity does not necessarily improve performance at each data scale. Furthermore, the overall performance on the ID test sets is generally much lower than on OOD (Fig. 2), where training on 100k diverse samples can recover up to 80% performance of $\mathcal{M}_{\text{ref}}$. These results quantitatively confirm that scale complements diversity by enhancing in-distribution performance, aligning with prior observations in narrower task setups or with task-specific measures [70, 72, 32].

### 2.3.2 Understanding What G-Vendi Encodes

Despite the surprising effectiveness of G-Vendi, it remains unclear whether the result stems from a fundamental difference in how G-Vendi defines diversity compared to baseline measures, such as *Embedding Vendi*. We analyze this by generating two specialized datasets, *DiverseReason* and *DiversePersona*. Specifically, we adopt the persona-guided generation framework of Ge et al. [9], and prompt Qwen2.5-72B-Instruct to generate math problems conditioned on both the few-shot examples and a random-sampled persona. For *DiverseReason*, we create 100k data points from 10k seed samples, along with only 0.1k personas. Conversely, for *DiversePersona*, we create 100k data points from only 0.1k seed samples, but with 10k personas. This design ensures that we induce a systematic difference between the two datasets, specifically by making *DiverseReason* richer in reasoning patterns but narrower in topical breadth and phrasing than *DiversePersona*.

Table 2 presents the diversity scores of the two datasets and the model performance after training on each dataset. As expected, the model trained on *DiverseReason* shows substantially better performance than with *DiversePersona*, as it has been exposed to a variety of problems with distinct reasoning patterns. Here, we find a notable difference between G-Vendi and *Embedding Vendi*—while G-Vendi consistently assigns higher score to subsets of *DiverseReason*, *Embedding Vendi*

Table 2: **Embedding diversity prioritizes semantic variability, while G-Vendi prioritizes solution template diversity.** We generate *DiverseReason* and *DiversePersona* in a controlled setup, and report their OOD performance and diversity scores along with standard deviation over 5 independent runs. See §F for qualitative examples comparing G-Vendi and *Embedding Vendi*.

| Dataset | N = 10k | | | N = 50k | | |
|---|---|---|---|---|---|---|
| | OOD Perf. | G-Vendi | Embedding Vendi | OOD Perf. | G-Vendi | Embedding Vendi |
| DiverseReason | **54.77 (3.49)** | **63.22 (1.05)** | 22.03 (0.89) | **63.48 (2.59)** | **103.63 (1.25)** | 30.79 (0.52) |
| DiversePersona | 38.15 (8.02) | 51.94 (1.24) | **24.49 (0.45)** | 42.14 (6.92) | 87.864 (1.19) | **31.16 (0.59)** |

gives higher diversity to *DiversePersona*, prioritizing semantic diversity over reasoning patterns. The results suggest that (1) heuristic diversification methods such as using persona may not necessarily improve the type of diversity helpful for the task, and (2) G-Vendi can be a useful tool to capture the task-specific notion of diversity—particularly in reasoning domains—by emphasizing variations in the underlying cognitive process rather than surface-level differences.

### 2.3.3 Impact of Gradient Proxy Model

Next, we ablate the proxy model used for computing gradient representations on math reasoning. In Table 3, we replace the proxy model with Llama-3.2-1B-Instruct and Qwen2.5-0.5B, and report their rank correlation on 10k subsets with (1) the original G-Vendi measured using Qwen2.5-0.5B-Instruct and (2) model performance. Llama3.2-1B-Instruct serves well as a proxy model, performing slightly better than Qwen2.5-0.5B-

Table 3: **G-Vendi is stable with proxy models of different sizes and model families.** We report rank correlation with the original proxy model and with model OOD performance on math reasoning.

| Proxy Model | $\rho$ w/ Qwen2.5-0.5B-It | $\rho$ w/ OOD Perf. |
|---|---|---|
| Qwen2.5-0.5B-It | 1 | 0.898 |
| Llama-3.2-1B-It | 0.899 | 0.909 |
| Qwen2.5-0.5B | 0.811 | 0.772 |

Instruct; we posit that this is because the model originates from the same base model as our student model $\mathcal{M}$ (base Llama3.2-1B). In addition, we find that using an instruction-tuned proxy is helpful, as the diversity measured under a base model yields a comparatively weaker correlation with model generalization. Overall, G-Vendi provides a stable estimate of data diversity with consistently high correlation across proxy models, suggesting its effectiveness is not specific to any single configuration.

### 2.3.4 Impact of Student Model

We also analyze whether G-Vendi is consistent across different student models other than Llama-3.2-1B. In Fig. 3 (Right), we randomly pick 5 distinct subsets from our math data pool, each with 10K samples, train 3 distinct student models on these subsets, then evaluate their OOD performance. Again, datasets with higher diversity consistently lead to better performance across model scales and families. Notably, Llama-3.1-8B [31] trained on the same-sized subsets show up to 25.4% difference in OOD accuracy, depending on the measured diversity of their training sets. The results show that G-Vendi captures a consistent aspect of dataset quality that holds across models—diverse datasets tend to be broadly beneficial, not just tailored to a particular training setup.

## 3 Prismatic Synthesis

The strong performance of G-Vendi provides us with a compelling prospect—by strategically improving data diversity in gradient space, we can improve our model performance **even without knowledge of the target distribution**. We present Prismatic Synthesis, a simple yet effective framework to improve data diversity by generating novel synthetic data. An overview of Prismatic Synthesis is shown in Fig 1. Starting from a seed dataset, the framework repeats the following 3 steps:

**Step 1: Cluster existing samples in gradient space.** Following §2.1, we compute the loss gradient of existing samples with an off-the-shelf proxy model, and cluster them using K-means.

**Step 2: Generate new samples from existing samples.** We prompt an LLM with few-shot examples sampled from the current data pool to generate new data points.

**Step 3: Diversify by leaving only the samples in sparse clusters.** Among the generated data points, we only add to the pool those samples that belong to sparse clusters (*e.g.,* the top 20% of clusters with the smallest number of members).

Iterating these steps, we collect the samples currently underrepresented in the gradient space, thereby greedily improving the data diversity. In the following section, we describe in detail how we apply Prismatic Synthesis to generate two state-of-the-art datasets, *PrismMath* and *PrismNLI*.

### 3.1 *PrismMath* and *PrismNLI*

**Generation and Diversification**   The generation process takes the few-shot approach in §2.2. For NLI, we start from WANLI [28] with 103k samples as seed dataset. We use Qwen2.5-72B-Instruct

Table 4: (Top) **PrismMath-7B, distilled from a 32B LLM, outperforms state-of-the-art reasoning models trained on solutions generated by a 671B LLM and further verified by human-written answers.** All scores are `pass@1`. (Bottom) **PrismNLI outperforms the best prior mixture of datasets by 8% across 8 OOD benchmarks.** We present full results with standard deviation and data size comparison in §B.2.

| Model | Data Generator | AIME24 | AIME25 | AMC23 | MATH500 | MATH$^2$ | Olympiad Bench | GSM8K Platinum | Avg. |
|---|---|---|---|---|---|---|---|---|---|
| Qwen2.5-Math-7B-It | - | 14.17 | 9.91 | 72.50 | 83.80 | 57.62 | 44.29 | 96.11 | 54.06 |
| OpenR1-Qwen-7B | | 47.91 | 30.41 | 87.19 | 90.60 | 78.10 | 67.06 | **96.69** | 71.14 |
| OpenThinker-7B | R1-671B | 27.50 | 22.50 | 74.06 | 84.20 | 67.62 | 45.93 | 93.05 | 59.27 |
| OpenThinker2-7B | | 50.00 | 35.00 | 88.44 | 91.40 | 78.10 | 69.63 | 93.96 | 72.36 |
| R1-7B | | 54.66 | 33.33 | 92.50 | **92.60** | 78.57 | 68.00 | 89.91 | 72.80 |
| **PrismMath-7B** | R1-32B | **57.08** | **38.33** | **93.75** | 92.40 | **80.95** | 68.30 | 95.95 | **75.25** |

| Dataset | Data Generator | HANS | WNLI | ANLI R1 | ANLI R2 | ANLI R3 | Diagnostics | BigBench | Control | Avg. |
|---|---|---|---|---|---|---|---|---|---|---|
| MNLI | | 78.47 | 63.03 | 60.10 | 45.30 | 42.58 | 81.88 | 78.22 | 42.16 | 61.47 |
| WANLI | ChatGPT, | 89.25 | 75.21 | 61.30 | 46.50 | 44.90 | 83.68 | 80.81 | 43.93 | 65.70 |
| MNLI+FEVER | Humans | 74.62 | 66.29 | 60.20 | 47.00 | 41.75 | 80.89 | 76.14 | 48.51 | 61.93 |
| {WA+M+S}NLI | | 80.18 | 69.41 | 65.00 | 50.60 | 45.25 | 83.77 | 84.72 | 50.49 | 66.18 |
| **PrismNLI** | Qwen2.5-72B | **92.44** | **78.47** | **73.70** | **61.90** | **57.00** | **86.13** | **86.32** | **58.25** | **74.28** |

to generate both the new problem and a corresponding label (either *entailment*, *neutral*, or *label*), given 5-shot examples random-sampled from the existing data pool. For math reasoning, we start from OpenR1-Math [16] with 94k samples. Instead of generating both problem and solution in one pass, we first generate problems by 5-shot prompting Qwen2.5-72B-Instruct. Then we use R1-Distill-Qwen-32B (R1-32B) to annotate solution traces for each generated problem. This two-stage process allows us to distill self-reflecting capabilities of R1-like models, known to be particularly effective in hard reasoning tasks [44, 64]. We dynamically set the number of clusters $k$ to be 1% of the existing data pool size at each iteration, and leave only the samples that correspond to the smallest $k/2$ clusters.

**Quality Filtering and Decontamination** Our pipeline generates data in an *entirely automated fashion*, i.e., it does not involve any ground-truth answers or manual verification. This is in contrast to dominant approaches for synthetic data generation, which first collect human-written problems and their ground-truth answers, then augment them with model-generated solution traces [23, 40]. To improve data quality, we perform majority-voting based filtering—we sample $N$ solutions for R1-32B for each generated problem, and compare their answers. When the number of majority answers is above a threshold $\tau$, we consider those answers to be "verified", and add them to the pool. We set $N = 3, \tau = 2$ for *PrismMath* and $N = 2, \tau = 2$ for *PrismNLI*. We then run decontamination against all test benchmarks we used (Table 4). We adopt the most conservative methods in literature, by first applying brute-force 10-gram matching and subsequently running LLM-based paraphrase detection [62]. After filtering and decontamination, we are left with 1.0M problem-solution pairs in *PrismMath*, and 515K input-label pairs for *PrismNLI*.

**Evaluation Setup** We evaluate the quality of our datasets by training models and evaluating them on a suite of benchmarks. For math reasoning, we fine-tune Qwen2.5-Math-7B-Instruct, yielding *PrismMath-7B*. We then compare our model against state-of-the-art distilled reasoning models at 7B scale, such as DeepSeek-R1-Distill-Qwen-7B (R1-7B) and OpenThinker2-7B. Note that these models are trained on heavily curated datasets, whose solution traces are generated by R1-671B and verified based on ground-truth answers. For NLI, we train Deberta-V3-large on PrismNLI, and compare against the same base model trained on (a mixture of) widely used datasets—such as a mixture of WANLI, MNLI and SNLI. We illustrate further experimental details in §A.2.

## 3.2 Evaluation Results

### 3.2.1 Main Results

**PrismMath-7B outperforms state-of-the-art distilled reasoning models.** The results for math reasoning tasks are shown in Table 4 (Top). *PrismMath-7B* yields surprisingly strong performance across benchmarks, outperforming all state-of-the-art baselines distilled for hard reasoning tasks. Notably, despite being distilled from R1-32B without any human verification involved, our model

outperforms OpenThinker2, a model trained on 1.14M samples distilled from R1-671B and further verified using ground-truth answers.

**PrismNLI improves by 8% from the best prior data mixture.** In Table 4 (Bottom), *PrismNLI* with 515k samples achieves substantially better OOD performance than a mixture of widely-used, large-scale datasets. For example, *PrismNLI* outperforms the mixture of WANLI, MNLI and SNLI by 8% on average OOD accuracy, despite being only the half the size of the baseline and not relying on any human annotation. Overall, these results show that the benefits of strategic diversification may exceed that of expensive curation techniques, such as a stronger data generator or manual verification. In §F, we analyze examples of actual clusters discovered in both math reasoning and NLI tasks.

### 3.2.2 Impact of Diversification on Synthetic Data Scaling

Is diversification in gradient space truly necessary? Although Prismatic Synthesis shows clear improvement in model performance, it is questionable whether similar gains could have been achieved with heuristic diversification—or even without any diversification, but by just scaling synthetic data. To address this concern, we compare the scaling of synthetic data generated from alternative diversification strategies—vanilla few-shot generation and persona-guided generation [9]—up to 100K samples. Further experimental details are in §A.2.

The results are shown in Fig. 1 (Right). In both vanilla few-shot and persona-guided generation, model performance faces early saturation—around $50K \sim 100K$ scale. Notably, while persona-guided generation outperforms other methods at low data scale (below 5K), it quickly plateaus, eventually converging to the performance achievable with vanilla few-shot prompting. This result attests to our earlier observation that heuristic diversification—that optimizes for variances in surface form—may ultimately fall short, as it overlooks the task-specific nature of diversity necessary for effective generalization. This contrasts with Prismatic Synthesis, where performance continues to improve even at scale beyond 100K, as demonstrated by PrismMath with 1 million samples (Table 4).

## 4 Related Work

Data diversity is often considered a key factor in LLM post-training. Motivating from ideas of data selection and active learning [35, 1, 21], prior works have shown that training on diverse instruction-tuning data improves sample efficiency and model robustness [72, 2, 30, 3, 11]. Yet, these approaches often rely on either task-specific heuristics—*e.g.,* variance in instruction metadata [58, 33, 65]—or embedding similarity [60, 66, 41] as a proxy for diversity, which may be insufficient for reasoning-oriented tasks. Gradient-based representation is a promising alternative to these proxies that can approximate training data influence [48], and has been adopted for data selection [68, 61]. Wang et al. [57] proposes a gradient kernel-based approach to instruction-tuning diversity, but requires LoRA adaptation and omits systematic comparison against canonical diversity measures. Our work builds upon these prior works, to (1) provide a large-scale empirical analysis on the impact of data diversity on model generalization, and (2) introduce G-Vendi, a scalable yet task-sensitive measure that requires no model adaptation.

Synthetic data are being increasingly adopted for improving LLM capabilities, particularly for reasoning-heavy domains such as math and code [44, 16, 34]. In these settings, LLMs often play a central role as data generators—augmenting solutions for existing prompts [40, 64], rephrasing human-curated datasets [65, 36], and bootstraping novel problems [54, 51]. Despite the broad applicability of synthetic data, subsequent analyses also report that naive scaling of synthetic data may yield significant duplicates and distributional biases [10, 67]. Several techniques have been proposed to improve data diversity—*e.g.,* conditioning on auxiliary attributes [67, 24, 9] or maximizing pairwise embedding distances [60]—but their efficacy essentially relies on the quality of the heuristic attributes and embeddings in capturing task-specific notion of diversity. Prismatic Synthesis provides a principled alternative to these approaches, allowing for improvements in model generalization through a task-agnostic gradient diversification process.

## 5 Conclusion

Our work investigates how to quantify and leverage data diversity to improve LLM reasoning. We show that the exponentiated entropy of data samples in a gradient space—as measured by G-Vendi—

strongly correlates with the empirical generalization of the model, significantly outperforming prior metrics that rely on heuristic features. Building on this insight, we introduce Prismatic Synthesis, an algorithm for targeted generation of diverse synthetic data in gradient space. Our resulting datasets, *PrismMath* and *PrismNLI*, yield state-of-the-art models that generalize well not only on in-distribution but also across challenging out-of-distribution benchmarks, highlighting the importance of principled diversification over strong generators and curation pipelines.

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

## A Experimental Details

### A.1 Evaluating Diversity Measures

**Generating Data Pool**   We further illustrate the experimental setup for evaluating data diversity measures. As described in §2.2, we 5-shot prompt Qwen2.5-72B-Instruct as data generator for both NLI and math reasoning tasks, generating 1.5M samples for both domains. For NLI, we generate both the input (*i.e.,* premise, hypothesis) and the corresponding label in one pass. For math reasoning, we take a two-step process, first generating novel problems given the few-shot examples, then solving each generated problem. Prompts for both tasks are shown in §A.3.

**Sampling Subsets**   We then sample distinct 300 subsets in total from the generated data pool, spanning a varied range of diversity. This process is non-trivial, however, because repeated random sampling of subsets would lead to subsets with similar diversity, thus not covering a wide spectrum of diversity values[2]. We therefore employ 4 distinct sampling strategies to cover the diversity spectrum:

- *Random Sampling:* Random-sample a subset of given size from the data pool.
- *Higher Diversity Sampling:* Search for a higher diversity subset with cluster-guided balanced sampling. Given representations of samples in the data pool, we first perform K-means clustering over the representations. We then perform balanced sampling across each cluster to up-sample from sparse clusters and down-sample from dense clusters in the given representation space.
- *Lower Diversity Sampling:* Reduce diversity by iteratively adding similar samples to the current members of the subset. Given representations of samples in the data pool, we randomly pick a small seed samples to initialize a subset, then iteratively add a batch of new samples whose similarity with at least one of the current subset members is above a predefined threshold.
- *Mixture Sampling:* Given a pair of random / high-diversity / low-diversity subsets, randomly mix them to create a new subset of a given size.

We present formal descriptions of *Higher Diversity Sampling* and *Lower Diversity Sampling* in Algorithm 1 and Algorithm 2. Note that both strategies do not search for globally maximal / minimal diversity, which distinguish them from submodular data selection methods [59, 38]; instead, we deliberately introduce adjustable parameters and stochasticity in to the algorithms so that we yield distinct subsets across diversity level, while not overlapping too much with each other. We use both the gradient and embedding representations of data, so that subsets are spread out not only with respect to G-Vendi but also in baseline metrics (*e.g., Embedding Vendi*).

**Training Models and Baseline Measures**   Finally, we train models on the sampled subsets. We train DeBERTa-v3-large and Llama-3.2-1B for our main experiments, which only requires 1 H100 GPU per training run thanks to the small number of parameters. In practice, we parallelize the training of distinct models over up to 8 H100 nodes. Along with G-Vendi, we consider 5 baseline measures widely used in the literature—*Embedding Vendi*, *Embedding Dissimilarity*, *2-gram Entropy*, *Average Perplexity*, *Skill Set Entropy*. *Average Perplexity* is measured with Qwen2.5-0.5B-Instruct, the same model as the gradient proxy model in G-Vendi. *Skill-Set Entropy* is an LLM-based approach where the taxonomy of *skill sets* are first collected by prompting LLMs with each sample in the data pool, then measuring the data diversity via the entropy of *skill sets* in the data points. Since we generate our data pool using MATH and GSM8k as seed set, we borrow the taxonomy of skill sets from Didolkar et al. [6] that extracted skill sets in MATH and GSM8k using GPT-4 [43]. We then prompt Qwen2.5-72B-Instruct to map each problem in our data pool to all corresponding skill sets from the taxonomy. Subsequently, we compute the entropy of skill sets in each subset.

### A.2 Prismatic Synthesis

**Generating *PrismMath* and *PrismNLI***   All datasets used for Prismatic Synthesis can be used for academic settings, licensed by either cc-by-4.0, MIT or Apache 2.0. The prompts shown for generating PrismMath and PrismNLI are shown in §A.3. For math reasoning, we generate solutions with max sequence length of 16K using R1-32B. All experiments and data generation are done in

---

[2]This is somewhat intuitive because if multiple subsets are sampled from the same population using the same sampling strategy, the resulting test statistics (e.g., the mean) are expected to exhibit low variance.

---

**Algorithm 1** *Higher Diversity Sampling*

---

**Input:** Data representation $D \in \mathbb{R}^{|\mathcal{D}| \times d}$, number of clusters $k$ and target subset size $N_{\text{target}}$
**Output:** Indices of selected subset $S \subseteq \{1, \cdots, |\mathcal{D}|\}$
   $S \leftarrow \emptyset$                    ▷ Initialize the subset.
   $\{c_1, \cdots, c_k\} = \text{K-means}(D)$    ▷ Cluster data. $c_i$ is a set of indices corresponding to cluster $i$.
   **while** $|S| < N_{\text{target}}$ **do**
      $c \leftarrow \text{random-sample}(\{c_1, \cdots, c_k\}, 1)$ ▷ Randomly pick a sampling cluster.
      $S_{\text{new}} \leftarrow \text{random-sample}\left(c, \left\lceil \frac{N_{\text{target}}}{k} \right\rceil\right)$ ▷ Sample new samples from the chosen cluster.
      $S \leftarrow S_{\text{new}}$                ▷ Add new samples to the subset.
   **return** $S$                  ▷ Return the sampled subset.

---

---

**Algorithm 2** *Lower Diversity Sampling*

---

**Input:** Data representation $D \in \mathbb{R}^{|\mathcal{D}| \times d}$, seed set size $N_{\text{seed}}$, batch size $N_{\text{batch}}$, target subset size $N_{\text{target}}$, similarity threshold $\tau$
**Output:** Indices of selected subset $S \subseteq \{1, \cdots, |\mathcal{D}|\}$
   $S \leftarrow \text{random-sample}(\{1, \cdots, |\mathcal{D}|\}$ ▷ Initialize the subset with seed data points.
   **while** $|S| < N_{\text{target}}$ **do**
      $S_{\text{new}} \leftarrow \{i \in \{1, \cdots |\mathcal{D}|\} \setminus S \mid \max_{j \in S} \text{cos-sim}(D_i, D_j) > \tau\}$
                                ▷ Find samples that are similar to the current subset members.
      $S_{\text{new}} \leftarrow \text{random-sample}(S_{\text{new}}, N_{\text{batch}})$
      $S \leftarrow S \cup S_{\text{new}}$          ▷ Add new samples to the subset.
   **return** $S$                  ▷ Return the sampled subset.

---

a local cluster without resorting to external APIs—since data generation is mostly parallelizable, we use 16 H100 nodes at maximum to expedite the generation process. We used 4 H100 nodes for fine-tuning models on our data. For decontamination, we first run brute-force 10-gram filtering, then perform LLM-based paraphrase detection following Yang et al. [62]. Specifically, we first match each generated sample with the closest sample in the evaluation benchmarks using an off-the-shelf embedding model, then run Qwen2.5-72B-Instruct to determine whether the generated sample is equivalent to the test sample. The paraphrase detection prompt can be found in §A.3.

**Evaluating *PrismMath* and *PrismNLI***   As shown in Table 4, we use HANS [37], WNLI, ANLI, Diagnostics [56], BigBench NLI [53] and ConTRoL [29] for NLI evaluation. For math reasoning, we use AIME24/25, AMC23, MATH500 [26], MATH$^2$ [51], Olympiad Bench [12] and GSM8k-Platinum [55]. Note here that we use a more challenging set of math benchmarks than in §2.2, since we aim to achieve state-of-the-art results on challenging benchmarks through long CoT reasoning, unlike in §2.2 where we primarily investigate the relative performance gap between models trained with distinct datasets. During evaluation, we generate solutions with temp $= 0.6$ and top-p $= 0.95$, and report `pass@1` for each benchmark. For AIME and AMC, we average `pass@1` over 16 independent runs to compensate for the small benchmark sizes. We use HuggingFace lighteval and math-verify [7] to evaluate answer correctness.

**Synthetic Data Scaling**   We analyze the impact of synthetic data scaling as shown in §3.2.2 and Fig. 1 (Right). We first generate a data pool of 100K samples using vanilla few-shot generation and persona-guided generation, respectively. For persona-guided generation, we random-sample personas from PersonaHub [9] and condition problem generation on each persona, along with 5 random-sampled few-shot examples from data pool (§A.3). After generating the data pool, we sample 5 distinct subsets (at each scale) from each data pool, train Qwen2.5-7B-Math on each dataset, and report the average test accuracy on the same set of benchmarks as above.

## A.3 Prompts

---

**Prompt for Math Problem Generation**

Given a set of example math problems, create a similar or harder problem inspired by the example problems.

The new problem should be formatted as:
——
[Problem]
your new problem - come up with a non-multiple choice problem, even if the provided examples are multiple choices.
——

Examples:
——
[Problem]
{example_problem_1}
——

...[few shot examples omitted]

---

**Prompt for Math Solution Generation**

Solve the following problem. Make sure to put your final answer in \boxed{}.

{input_problem}

---

**Prompt for Math Problem Generation with Persona**

Example 1:
{example_problem_1}

...[few shot examples omitted]

Create a challenging math problem similar to the examples above with the following persona:
{input_persona}

---

## Prompt for NLI Sample Generation

Given examples of Natural Language Inference task, create a novel and more challenging problem with the similar reasoning as the given examples.

Each of the novel example should be formatted as:
—
Premise: premise text
Hypothesis: hypothesis text
Label: label
—

Examples:
—
Premise: {example_premise_1}
Hypothesis: {example_hypothesis_1}
Label: {example_label_1}
—

...[few shot examples omitted]

## Prompt for NLI Sample Verification

Given a pair of premise and hypothesis, determine if the hypothesis is entailed by, or contradicted by, or neutral to the premise.
Your answer should be either 'Entailment', 'Contradiction' or 'Neutral'.

Premise: {input_premise}
Hypothesis: {input_hypothesis}
Label (Entailment or Contradiction or Neutral):

## Prompt for Contamination Detection

Given two problems, help me determine if the two problems are equivalent.
- Disregard the names and minor changes in word order that appear within.
- If they are equivalent, please answer 'True', otherwise answer 'False'. Do not respond with anything else.
- If their question prompts are very similar and, without considering the solution process, they produce the same answer, we consider them to be the same question.

Problem 1: {input_problem_1}
Problem 2: {input_problem_2}

# B Additional Results

## B.1 Results on Baseline Measures

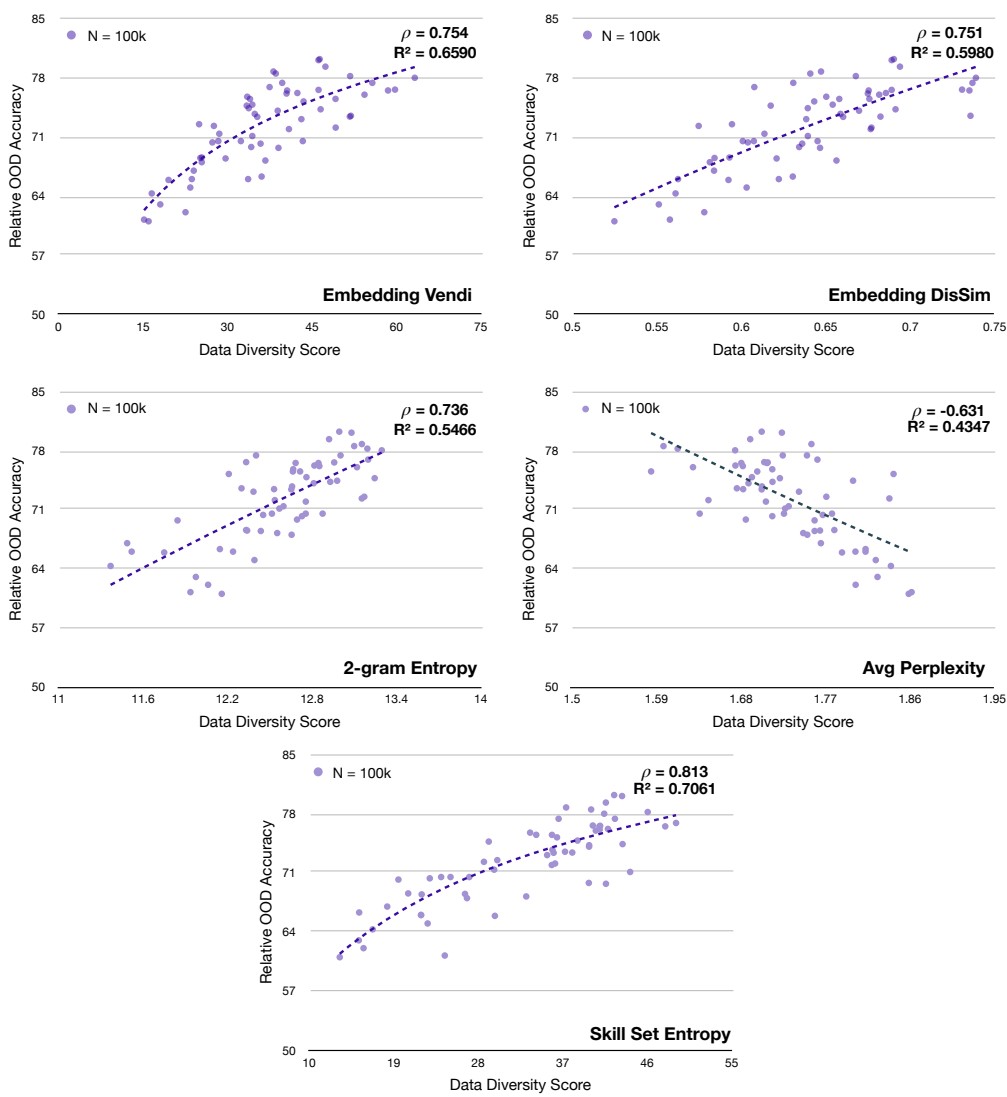

Figure 4: Relationship between baseline diversity measures and model OOD performance, measured in math reasoning tasks. Relative OOD accuracy is averaged across 7 benchmarks, following the same process as in §2.2. Overall, widely-used diversity measures fall behind G-Vendi in their correlation with model performance.

## B.2 Extended Results on Prismatic Synthesis

In Table 5 and 6, we present full results on *PrismMath* and *PrismNLI* including the standard error for AIME24 / AIME25 / AMC23 and dataset size comparison.

Table 5: Full evaluation results on math reasoning benchmarks, including dataset size comparison. All scores are pass@1, and for AIME and AMC, we report average pass@1 across 16 runs, and report their standard error.

| Model | Data Generator | Data Size | AIME24 | AIME25 | AMC23 | MATH500 | MATH$^2$ | Olympiad Bench | GSM8K Platinum | Avg. |
|---|---|---|---|---|---|---|---|---|---|---|
| Qwen2.5-Math-7B-It | - | - | 14.17 (0.83) | 9.91 (1.06) | 72.50 (1.68) | 83.80 | 57.62 | 44.29 | 96.11 | 54.06 |
| OpenR1-Qwen-7B | | 114K | 47.91 (1.88) | 30.41 (1.72) | 87.19 (1.92) | 90.60 | 78.10 | 67.06 | **96.69** | 71.14 |
| OpenThinker-7B | R1-671B | 1.14M | 27.50 (1.22) | 22.50 (1.97) | 74.06 (1.15) | 84.20 | 67.62 | 45.93 | 93.05 | 59.27 |
| OpenThinker2-7B | | 94K | 50.00 (1.67) | 35.00 (2.44) | 88.44 (1.83) | 91.40 | 78.10 | **69.63** | 93.96 | 72.36 |
| R1-Distill-7B | | Unknown | 54.66 (1.96) | 33.33 (1.52) | 92.50 (1.24) | **92.60** | 78.57 | 68.00 | 89.91 | 72.80 |
| **PrismMath-7B** | R1-32B | 1.0M | **57.08** (1.93) | **38.33** (1.41) | **93.75** (1.06) | 92.40 | **80.95** | 68.30 | 95.95 | **75.25** |

Table 6: Full evaluation results on NLI benchmarks, including dataset size comparison.

| Dataset | Data Generator | Data Size | HANS | WNLI | ANLI R1 | ANLI R2 | ANLI R3 | Diagnostics | BigBench | Control | Avg. |
|---|---|---|---|---|---|---|---|---|---|---|---|
| MNLI | | 393K | 78.47 | 63.03 | 60.10 | 45.30 | 42.58 | 81.88 | 78.22 | 42.16 | 61.47 |
| WANLI | ChatGPT, | 103K | 89.25 | 75.21 | 61.30 | 46.50 | 44.90 | 83.68 | 80.81 | 43.93 | 65.70 |
| MNLI+FEVER | Humans | 601K | 74.62 | 66.29 | 60.20 | 47.00 | 41.75 | 80.89 | 76.14 | 48.51 | 61.93 |
| {WA+M+S}NLI | | 943K | 80.18 | 69.41 | 65.00 | 50.60 | 45.25 | 83.77 | 84.72 | 50.49 | 66.18 |
| **PrismNLI** | Qwen2.5-72B | 515K | **92.44** | **78.47** | **73.70** | **61.90** | **57.00** | **86.13** | **86.32** | **58.25** | **74.28** |

## C  Properties of G-Vendi

As the aggregation process of G-Vendi follows that of Vendi Score [8], it inherits several desirable properties of Vendi score as a diversity metric. More specifically,

- G-Vendi is positive unbounded, and can be interpreted as the effective number of unique samples in a given dataset. That is, if the normalized covariance matrix $K$ of dataset $\mathcal{D}$ satisfies $K_{ij} = 0$ for all $i \neq j \in [1, |\mathcal{D}|]$, then G-Vendi$(\mathcal{D}) = |\mathcal{D}|$. If $K_{ij} = 1$ for all $i \neq j \in [1, |\mathcal{D}|]$, G-Vendi$(\mathcal{D}) = 1$.
- G-Vendi is permutation invariant. That is, permuting the order of data points in $\mathcal{D}$ does not change the value of G-Vendi$(\mathcal{D})$.

In terms of computational complexity, computing eigenvalues would cost $O(n^3)$ for an $n \times n$ matrix [47]. This is prohibitively inefficient to directly compute for our normalized covariance matrix $K$, since $K = 1/|\mathcal{D}| \cdot GG^T \in \mathbb{R}^{|\mathcal{D}| \times |\mathcal{D}|}$ for the data representation matrix $G \in \mathbb{R}^{|\mathcal{D}| \times d}$. However, we can leverage the fact that eigenvalues of $K = 1/|\mathcal{D}| \cdot GG^T$ equals that of $1/|\mathcal{D}| \cdot G^T G \in \mathbb{R}^{|d| \times |d|}$. Therefore, the time complexity for G-Vendi involving both the computation of $G^T G$ and the eigenvalues of $G^T G$ is $O(d^3 + d^2|\mathcal{D}|) = O(d^2|\mathcal{D}|)$. This is much more efficient than the average pairwise similarity that requires the actual similarity matrix $GG^T$, which takes $O(d|\mathcal{D}|^2)$.

## D  Limitations and Future Works

One of our main goals in this work is to investigate the impact of data diversity on the trained model's behavior. While our analyses reveal a strong correlation between G-Vendi and the generalization of the trained model to diverse benchmarks, G-Vendi is not a panacea—the strong correlation can only be discovered via careful control of data quality. Thus, G-Vendi cannot be used as a sole metric for an apple-to-apple comparison between two arbitrary datasets (that originated from completely different data generating processes). Analyzing the interplay between the details of data-generating process (*e.g.,* verification protocol or quality filtering) and data diversity, and their impact on the model performance remains an interesting future direction.

In addition, our measure of model generalization relies on aggregating performance across multiple benchmarks. While aggregation over diverse benchmarks is indeed a standard setup for evaluating LLM capabilities, it may obfuscate the impact of data diversity on a new domain, where not many off-the-shelf benchmarks exist. An interesting future work motivated by this gap, is to compute *target-conditional data diversity*—which not only considers the intrinsic diversity of the training dataset itself, but also the relevance to the target distribution we want to excel on. Incorporating with prior literature on targeted instruction tuning [61], Prismatic Synthesis could evolve to generate diverse synthetic data that are targeted specifically to improve model capabilities in a given benchmark.

## E  Broader Impacts

Our work contributes to a more principled understanding of data diversity in LLM reasoning, with potential positive implications for building models that generalize better by training on diverse inputs. By proposing scalable methods for measuring and generating cognitively diverse data, our approach could inform better dataset construction practices and reduce overreliance on large, opaque corpora. However, improvements in synthetic data generation may also lower the barrier to creating high-performing models for malicious use, such as for generating more convincing misinformation or deceptive content. Care should be taken to ensure these methods are applied in alignment with ethical guidelines and responsible model deployment.

# F  Qualitative Examples

## F.1  Comparing Closet Sample in Gradient Space and Embedding Space

In Example 1 to 3 below, we qualitatively compare the most similar data point to a given math reasoning sample, measured in the gradient space and the embedding space. We use the same setup as in §2.2, and select GSM8k problems that are easier to understand.

Overall, we find that the closest sample in embedding space often shares the central theme or keywords with the original sample (*e.g.,* a figure collecting Pokemon cards in Example 1). This contrasts with the sample closest in the gradient space, which often does not exhibit any semantic relevance to the original problem, but often shares a very similar solution strategy with the seed sample. These examples qualitatively indicate that similarity in gradient space can better capture the task-specific cognitive process for problem solving.

---

**Example 1**

**Original Sample**

***Problem***: Jenny has 6 Pokemon cards. Orlando has 2 more cards than Jenny. Richard has three times as many cards as Orlando. How many Pokemon cards do they have in all?

***Solution***: To solve the problem step by step, we start with the information given:
1. Jenny has 6 cards. This is given directly in the problem statement. ... [omitted]

Therefore, the total number of Pokemon cards they have all together is $\boxed{38}$ cards.

- - - - - - - - - - - - - - - - - - - - - - - - - - - - - - - - - - - - - - - - - - - - - - -

**Closest Sample in Gradient Space**

***Problem***: Levi has 5 lemons. Jayden has 6 more lemons than Levi. Jayden has one-third as many lemons as Eli has while Eli has one-half as many lemons as Ian has. How many lemons do they have in all?

***Solution***: To solve the problem step by step, we start with the given information and work our way through the relationships between the number of lemons each person has. ... [omitted]

Therefore, the total number of lemons they all have is $\boxed{115}$ lemons.

- - - - - - - - - - - - - - - - - - - - - - - - - - - - - - - - - - - - - - - - - - - - - - -

**Closet Sample in Embedding Space**

***Problem***: Nicole collected 400 Pokemon cards. Cindy collected twice as many, and Rex collected half of Nicole and Cindy's combined total. If Rex divided his card equally among himself and his three younger siblings, how many cards does Rex have left?

***Solution***: To solve this problem, we follow the steps based on the information provided:
1. Calculate the number of cards Cindy has: ... [omitted]

So, the number of cards Rex has left is $\boxed{150}$.

---

## Example 2

**Original Sample**

***Problem***: The number of math problems that Marvin practiced today is three times as many as the number of problems he solved yesterday. His friend, Arvin, has practiced twice as many math problems on each day. How many math problems have they practiced altogether if Marvin solved 40 math problems yesterday?

***Solution***: Given that Marvin solved 40 math problems yesterday, we can calculate the number of problems he solved today and the total number of problems solved by both Marvin and Arvin as follows: ...[omitted]

Therefore, together, Marvin and Arvin have solved $\boxed{480}$ math problems.

- - - - - - - - - - - - - - - - - - - - - - - - - - - - - - - - - - - - - - - -

**Closest Sample in Gradient Space**

***Problem***: Olaf has an aquarium. He has fish in 3 different colors: orange, green, and blue. Blue fish make up half of all the fish in the aquarium. There are 15 fewer orange fish than blue fish. How many green fish are there when the total number of fish in the aquarium is 80?

***Solution***: Given the total number of fish in the aquarium is 80, and blue fish make up half of all the fish, we can calculate the number of blue fish as follows: ...[omitted]

Therefore, the number of green fish in the aquarium is $\boxed{15}$.

- - - - - - - - - - - - - - - - - - - - - - - - - - - - - - - - - - - - - - - -

**Closet Sample in Embedding Space**

***Problem***: Angela wants to check her math homework answers with her friends, but some of them aren't done yet. Out of 20 problems, Martha has finished 2, JJenna has finished four times the number Martha did minus 2, and Mark has finished half the number Jenna did. If none of the friends worked on any of the same problems, how many problems have no one but Angela finished?

***Solution***: To solve the problem, let's break down the solution step by step:

1. **Calculate the number of problems Jenna answered:** ...[omitted]

Therefore, Angela was able to finish $\boxed{9}$ problems on her own.

## Example 3

**Original Sample**

*Problem*: The $4.55 in Carol's piggy bank consists of quarters and nickels. There are seven more nickels than quarters. How many nickels does Carol have in her bank?

*Solution*: To solve the problem, we start by defining variables for the number of nickels and quarters Carol has. Let $n$ represent the number of nickels and $q$ represent the number of quarters. Given that . . . [omitted]

Therefore, Carol has $\boxed{21}$ nickels in her piggy bank.

- - - - - - - - - - - - - - - - - - - - - - - - - - - - - - - - - - - - - - - - - - - - -

**Closest Sample in Gradient Space**

*Problem*: Billy Goats invested some money in stocks and bonds. The total amount he invested was $165,000. If he invested 4.5 times as much in stocks as he did in bonds, what was his total investment in stocks?

*Solution*: To solve the problem, we start by letting the amount Billy Goats invested in bonds be $s$. Since he invested 4.5 times as much in stocks, . . . [omitted]

Therefore, the total investment in stocks is $\boxed{\$135,000}$.

- - - - - - - - - - - - - - - - - - - - - - - - - - - - - - - - - - - - - - - - - - - - -

**Closet Sample in Embedding Space**

*Problem*: Maria has 4 dimes, 4 quarters, and 7 nickels in her piggy bank. Her mom gives her 5 quarters. How much money, in dollars, does Maria have now?

*Solution*: To calculate the total amount of money Maria has in her piggy bank after her mom gives her additional quarters, we proceed as follows:

1. **Calculate the total number of quarters Maria has now:**. . . [omitted]

Therefore, the total amount of money Maria has in her piggy bank is $\boxed{\$3.00}$.

## F.2 Example Clusters in Gradient Space

We additionally analyze clusters of data points in gradient space, for both NLI and math reasoning tasks. These examples are specifically selected for interpretability, as many gradient clusters are difficult to analyze. However, we find that when interpretable, the cluster members often exhibit a common, nuanced reasoning strategies rather than superficial topical or semantic overlaps.

---

**NLI Example 1: Membership between concepts**

*Premise*: James Blake ... [omitted] bus driver, ordered a 42-year old woman to move further back on the bus and to give her seat to a white person. When she did not comply, he called the police and 4 of them came on board. They arrested Rosa Parks who refused to move further back, not because the bus was crowded but because **the front 4 rows** were reserved for white people.

*Hypothesis*: Rosa Parks was seated in **one of the 4 front rows** of the bus.

*Label*: Entailment

- - - - - - - - - - - - - - - - - - - - - - - - - - - - - - - - - - - - - - - - - - - - -

*Premise*: The Australian Open, French Open, Wimbledon, and US Open are the 4 most prestigious tennis tournaments in the world. Winning all four in the same calendar year is considered a Grand Slam.

*Hypothesis*: If a player wins the **French Open and the other three Grand Slam tournaments**, they will have won *all four* Grand Slam tournaments.

*Label*: Entailment

- - - - - - - - - - - - - - - - - - - - - - - - - - - - - - - - - - - - - - - - - - - - -

*Premise*: American Pharoah won the Triple Crown in 2015, becoming the first horse to do so since Affirmed in 1978. *The Triple Crown consists of the Kentucky Derby, the Preakness Stakes, and the Belmont Stakes.* To win the Triple Crown, a horse must win all 3 races in a single season.

*Hypothesis*: American Pharoah won the **Kentucky Derby** in 2015.

*Label*: Entailment

---

**NLI Example 2: Traits of a person**

*Premise*: Henk Fraser is a Dutch former football player and coach. He is the sone of famous Dutch football player, Bert Fraser. Henk coached Sparta Rotterdam, ADO Den Haag and assisted on Ajaex A1. Now he is an assistant coach at Vittesse.

*Hypothesis*: Henk Fraser **is in** sports industry.

*Label*: Entailment

- - - - - - - - - - - - - - - - - - - - - - - - - - - - - - - - - - - - - - - - - - - - -

*Premise*: The television series "Smallville" revolves around Clark Kent and his friends as they try to navigate their lives and stop various villains in the fictional town of Smallville, Kansas.

*Hypothesis*: Clark Kent **lives in** the state of Kansas.

*Label*: Entailment

- - - - - - - - - - - - - - - - - - - - - - - - - - - - - - - - - - - - - - - - - - - - -

*Premise*: Wandi Rum is a village in southerm Jordan, southeast of the city Aqaba. It is set in an extraordinary landscape of mountains, valleys, dunes and Bedouin camps.

*Hypothesis*: Wadi Rum **has** Bedouin residents.

*Label*: Entailment

NLI Example 3: Key information at the beginning of premise

*Premise*: The American Broadcasting Company **(ABC) is a major American commercial broadcast network**, and the fifth-oldest major network in the US. It broadcasts a wide range of programming, including drama television series, which has become increasingly popular in [...omitted] over the past two decades.
*Hypothesis*: ABC **broadcasts American** drama television series.

*Label*: Entailment

- - - - - - - - - - - - - - - - - - - - - - - - - - - - - - - - - - - - - - - - - - -

*Premise*: **The store's advertisement for a Halloween sale** features a giant spider hanging from the celing and mannequins dressed in elaborate costumes and masks.
*Hypothesis*: The store sells **Halloween decorations and accessories**.

*Label*: Entailment

- - - - - - - - - - - - - - - - - - - - - - - - - - - - - - - - - - - - - - - - - - -

*Premise*: **Liam believes that several people in a gaming arcade are sitting at computes** while wearing headphones and intensely focused on the screens.
*Hypothesis*: **Liam believes that people** play computer games in the arcade.

*Label*: Entailment

NLI Example 4: Aggregating "different" concepts

*Premise*: The Asian Games were held in Bangkok in **1973**, and in Hiroshima in **1978**.
*Hypothesis*: The Asian Games were held in **different years**.

*Label*: Entailment

- - - - - - - - - - - - - - - - - - - - - - - - - - - - - - - - - - - - - - - - - - -

*Premise*: The 2024 European Women's Handball Championship will be the 16th edition of European Woman's Handball Championship, the top level woman's handball event organized by the European Handball Federation. The tournament will take place in **Austria, Hungary and Switzerland**.
*Hypothesis*: The European Women's Handball Championship will be held in **three different countries**.

*Label*: Entailment

- - - - - - - - - - - - - - - - - - - - - - - - - - - - - - - - - - - - - - - - - - -

*Premise*: For the past century the city has been **dominated by the German and French**, and in the past few years by the British.
*Hypothesis*: The city has been dominated by **different people**.

*Label*: Entailment

**Problem**: Simon, Gerry, and Micky want to have a race with handmade miniature rafts. Simon's raft needs 36 sticks, Gerry's raft needs two-thirds of the number of sticks that Simon needs, and Micky's raft needs 9 sticks more than Simon and Gerry's rafts combined. How many sticks do the three boys need to collect in total?

**Solution**: To solve this problem, we follow the information given step by step:

1. **Simon's Raft**: Simon needs 36 sticks. ... [omitted]

Therefore, the total number of sticks the three boys need to collect is $\boxed{129}$ sticks.

- - - - - - - - - - - - - - - - - - - - - - - - - - - - - - - - - - - -

**Problem**: Bella bought stamps at the post office. Some of the stamps had a snowflake design, some had a truck design, and some had a rose design. Bella bought 11 snowflake stamps. She bought 9 more truck stamps than snowflake stamps, and 13 fewer rose stamps than truck stamps. How many stamps did Bella buy in all?

**Solution**: To solve the problem, we follow the information given step by step:

1. Bella bought 11 snowflake stamps. ... [omitted]

Thus, the total number of stamps Bella bought is $\boxed{38}$.

- - - - - - - - - - - - - - - - - - - - - - - - - - - - - - - - - - - -

**Problem**: Marcy is the lunch monitor in an elementary school cafeteria. She gives 5 time-outs for running, 1 less than five times that number of time-outs for throwing food, and 1/3 the number of food-throwing time-outs for swearing. If each time-out is 5 minutes, how much time do the students spend in time-out total?

**Solution**: To solve the problem step by step step, we follow the given instructions closely:

1. **Calculate the number of time-outs for running**: ... [omitted]

Thus, the students spend a total of $\boxed{185}$ minutes in time-out.

- - - - - - - - - - - - - - - - - - - - - - - - - - - - - - - - - - - -

**Problem**: Rebecca makes her own earrings out of buttons, magnets, and gemstones. For every earring, she uses two magnets, half as many buttons as magnets, and three times as many gemstones as buttons. If Rebecca wants to make 4 sets of earrings, how many gemstones will she need?

**Solution**: To solve the problem, we follow the information given and calculate step by step:
1. **Calculate the total number of earrings**: ... [omitted]

Therefore, to make 4 sets of earrings, Rebecca will need $\boxed{24}$ gemstones.

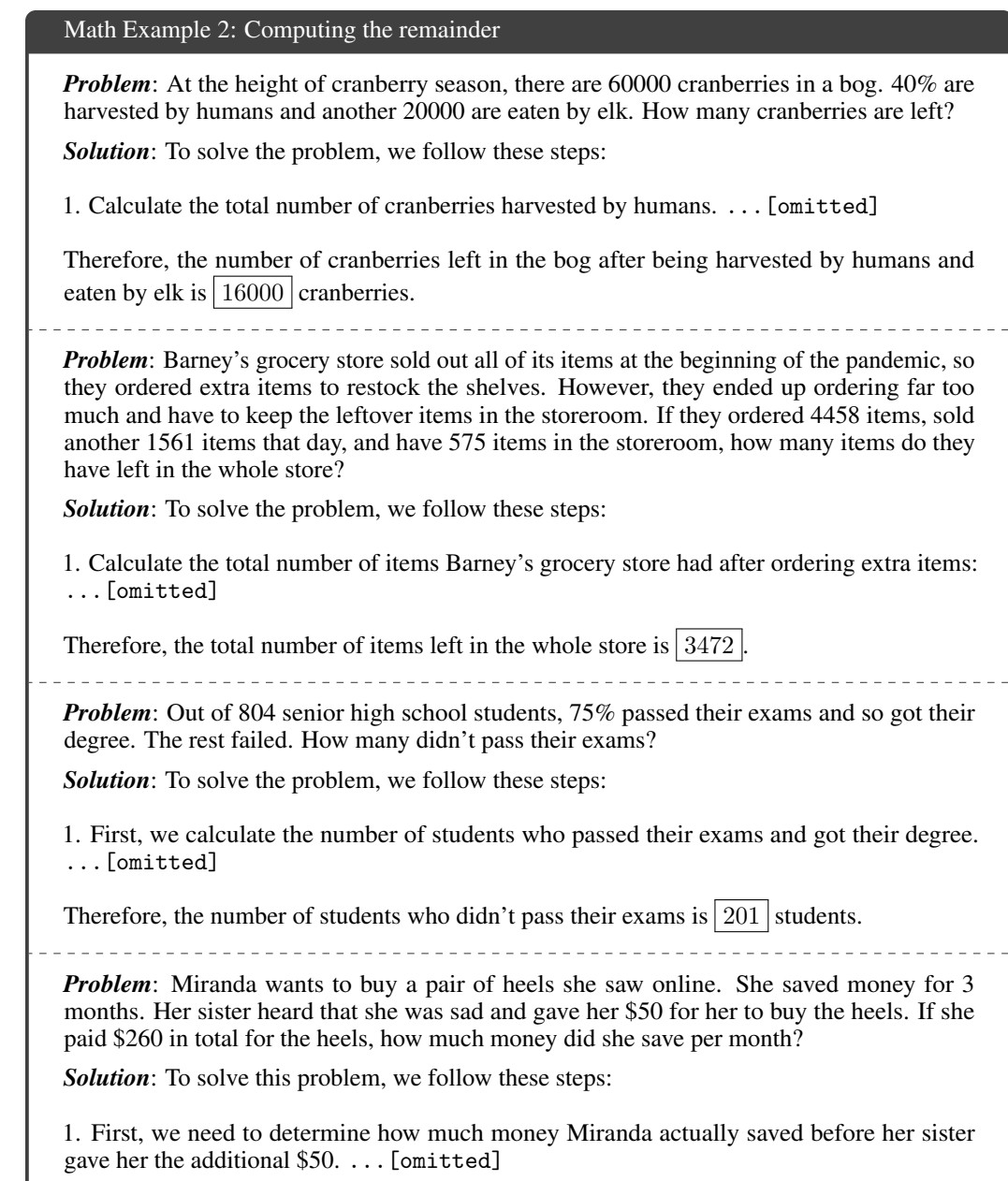

Math Example 2: Computing the remainder

**Problem**: At the height of cranberry season, there are 60000 cranberries in a bog. 40% are harvested by humans and another 20000 are eaten by elk. How many cranberries are left?

**Solution**: To solve the problem, we follow these steps:

1. Calculate the total number of cranberries harvested by humans. ...[omitted]

Therefore, the number of cranberries left in the bog after being harvested by humans and eaten by elk is 16000 cranberries.

- - - - - - - - - - - - - - - - - - - - - - - - - - - - - - - - - - - - - - - -

**Problem**: Barney's grocery store sold out all of its items at the beginning of the pandemic, so they ordered extra items to restock the shelves. However, they ended up ordering far too much and have to keep the leftover items in the storeroom. If they ordered 4458 items, sold another 1561 items that day, and have 575 items in the storeroom, how many items do they have left in the whole store?

**Solution**: To solve the problem, we follow these steps:

1. Calculate the total number of items Barney's grocery store had after ordering extra items: ...[omitted]

Therefore, the total number of items left in the whole store is 3472 .

- - - - - - - - - - - - - - - - - - - - - - - - - - - - - - - - - - - - - - - -

**Problem**: Out of 804 senior high school students, 75% passed their exams and so got their degree. The rest failed. How many didn't pass their exams?

**Solution**: To solve the problem, we follow these steps:

1. First, we calculate the number of students who passed their exams and got their degree. ...[omitted]

Therefore, the number of students who didn't pass their exams is 201 students.

- - - - - - - - - - - - - - - - - - - - - - - - - - - - - - - - - - - - - - - -

**Problem**: Miranda wants to buy a pair of heels she saw online. She saved money for 3 months. Her sister heard that she was sad and gave her $50 for her to buy the heels. If she paid $260 in total for the heels, how much money did she save per month?

**Solution**: To solve this problem, we follow these steps:

1. First, we need to determine how much money Miranda actually saved before her sister gave her the additional $50. ...[omitted]

Therefore, Miranda saved $70 per month.

## Math Example 3: How old is $X$ now?

***Problem***: In 10 years, Cornelia will be three times as old as Kilee. If Kilee is currently 20 years old, how old is Cornelia now?

***Solution***: To solve this problem, we follow these steps:

1. Calculate Kilee's age in 10 years: ... [omitted]

Therefore, Cornelia is currently $\boxed{80}$ years old.

- - - - - - - - - - - - - - - - - - - - - - - - - - - - - - - - - - - - - - - - - -

***Problem***: In 5 years, Raven will be 4 times as old as Phoebe. If Phoebe is currently 10 years old, how old is Raven?

***Solution***: To solve this problem, we follow these steps:

1. Calculate Phoebe's age in 5 years:... [omitted]

Therefore, Raven is currently $\boxed{55}$ years old.

- - - - - - - - - - - - - - - - - - - - - - - - - - - - - - - - - - - - - - - - - -

***Problem***: After five years, Ron will be four times as old as Maurice. If Ron's age now is 43, how old is Maurice now?

***Solution***: 1. Calculate Ron's age in five years:
- Ron's current age is 43 years.... [omitted]

Therefore, Maurice is currently $\boxed{7}$ years old.

- - - - - - - - - - - - - - - - - - - - - - - - - - - - - - - - - - - - - - - - - -

***Problem***: In four years, Suzy will be twice Mary's age then. If Suzy is 20 now, how old is Mary?

***Solution***: To solve this problem, we follow these steps:
1. Calculate Suzy's age in four years: ... [omitted]

Thus, Mary is $\boxed{8}$ years old now.

