# OpenReview forum: "Prismatic Synthesis: Gradient-based Data Diversification Boosts Generalization in LLM Reasoning"
_NeurIPS.cc/2025/Conference — NeurIPS 2025 spotlight_

### Official Review · Reviewer_3sq9 · 2025-07-01

**Clarity:** 4
**Significance:** 3
**Originality:** 3
**Rating:** 5
**Confidence:** 2

**Summary:**

This paper proposes to measure the "diversity" of a dataset used to train a model by measuring the entropy of a proxy model's gradients on this data distribution. The paper shows that this measure outperforms other metrics (like embedding diversity), and shows how it can be used to guide synthetic data sampling for model training.

**Questions:**

1. Is the gradient entropy measure useful only from "base" models, or is it a useful metric, say during training? E.g. what does the gradient entropy for a dataset look like using model iterates from fine-tuning?
2. What is the relative cost of computing the gradient entropy? Given a batch of $bs \times seq\_len$ tokens, what is the cost of computing the gradient entropy vs computing the avg minibatch gradient?

**Ethical Concerns:**

["NO or VERY MINOR ethics concerns only"]

**Final Justification:**

The response satisfactorily addresses my concerns, and I am keeping my score of accept.

**Quality:**

3

**Strengths And Weaknesses:**

- The idea of using gradient diversity is a nice and conceptually well-motivated idea for measuring dataset diversity, and the use of random matrix projection to make this computation tractable enables this to be done in a reasonably scalable manner
- Various aspects of the measure (e.g. the choice of proxy model, how it compares to various downstream metrics) are well analyzed
- This metric seems to be better than other approaches (like embedding or perplexity-based metrics) in predicting utility
- I thought the use of Diverse-{Reason, Persona} to disentangle what diversity from embedding vs gradient diversity looks like to be didactic
- The paper is well-written and easy to follow.

Weaknesses:

For the comparisons in Table 4, it would have been useful to compare the G-Vendi approach to performing data generation with other diversity metrics that were measured in prior sections (like embedding diversity, perplexity, etc). As it stands right now, it is hard to draw conclusions since the data generation is not controlled for between Prism-{NLI, Math} and the other comparisons.

---

> ### Author Rebuttal · Authors · 2025-07-30
>
> We sincerely thank the reviewer for the positive feedback and constructive questions! We are glad that you find our idea to be nice and well-motivated, our analyses to be thorough, our work to be empirically strong and well-written! Below, we address each of your questions:
>
> ## Controlled baselines for data generation?
> We appreciate the reviewer’s concern. We would like to gently refer to Section 3.2.2, where we ask *whether diversification in gradient space is truly necessary* — conducting controlled experiments via naive scaling & heuristic diversification under the same condition as in Prismatic Synthesis. In addition to the existing results, here we add an additional baseline that leverages embedding diversity. In the same controlled setup as in our main setup, we leverage embedding-based clustering instead of gradients in Prismatic Synthesis:
>
> | Dataset Generation Method     | Avg Test Acc |
> |-------------------------------|--------------|
> | Vanilla few-shot generation   | 57.93        |
> | Persona-guided generation     | 60.35        |
> | Embedding Prismatic Synthesis | 57.82        |
> | Prismatic Synthesis           | 61.72        |
>
> The results at 10k scale is shown above. We find that Embedding-based diversification, just like the other baselines from Section 3.2.2, does not lead to improvements comparable to Prismatic Synthesis. Overall, these results demonstrate the high utility of gradient information in capturing diverse reasoning patterns.
>
> ## Prospect of “online proxy model”
> We sincerely appreciate this great suggestion! In this work we did focus on a fixed proxy model, in particular because even the impact of diversity on a static dataset has not been widely understood. But updating the proxy model throughout the course of model training would be an interesting and reasonable next step to explore. We believe this would holistically improve the relevance of gradient information we get through the proxy model, particularly in an active learning setup (e.g. dynamically selecting diverse subsamples during training).
>
> ## Relative cost of gradient entropy
> As expected, much of the operation we perform in G-Vendi computation is similar to canonical gradient descent. For example, we do forward propagation once as in the typical mini-batch gradient descent. The only difference is that for G-Vendi, we backpropagate for each sample loss individually, as we want to represent each sample with their respective gradient vectors. Other overheads such as computing the entropy from collected gradient features are constant, thus is negligible compared to the gradient collection stage.
>
> We thank the reviewer again for thoughtful feedback!

---

### Official Review · Reviewer_94F8 · 2025-07-03

**Clarity:** 3
**Significance:** 4
**Originality:** 3
**Rating:** 5
**Confidence:** 2

**Summary:**

This paper discussed measurement of data diversity and proposes a method to amplify it. First, it introduces G-Vendi, a metric that measures the diversity of a given dataset. Moreover, it proposes Prismatic Synthesis that can synthesize new data to expand the diversity. Experiments demonstrate the improvements.

**Questions:**

See the weakneeses.

**Ethical Concerns:**

["NO or VERY MINOR ethics concerns only"]

**Limitations:**

yes

**Quality:**

3

**Strengths And Weaknesses:**

Strengths:
1. It explores the measurement of data diversity, and first proposes the metric (i.e., G-Vendi) that precisely measures the diversity of a given dataset. This could be significantly beneficial for collecting data and data postprocessing.
2. Experiments are good and extensive, and the compared counterparts are sufficient.
3. The overall presentation and writing are clear.

Weaknesses:
1. The comparison in experiments in Table 2 is sparse. The gap between DiverseReason and DiversePersona is too large to prove that G-Vendi is a sensitive metric. Maybe an intermediate dataset should be added in this comparison.
2. Experiments in Table 3 are based on small models (<= 1B). How about larger ones (>= 7B)?

---

> ### Author Rebuttal · Authors · 2025-07-30
>
> We thank the reviewer for both the positive and thoughtful comments! We are happy that you find our work to be of significant benefit for LLM data curation, our experiments to be well-designed, and the overall presentation to be clear. Below, we address each of your concerns.
>
> ## Intermediate dataset between DiverseReason and DiversePersona
>
> | Dataset        | OOD Perf. | G-Vendi | Embedding Vendi |
> |----------------|-----------|---------|-----------------|
> | DiverseReason  | 54.77     | 63.22   | 22.03           |
> | Mixture        | 44.27     | 58.01   | 24.43           |
> | DiversePersona | 38.15     | 51.94   | 24.49           |
>
> We agree that adding another dataset in between the two extremes could further strengthen the findings of our analyses. Instead of adding a heuristic pipeline that may lie in the middle of the two datasets, we take a straightforward approach of taking 50%-50% mixture of the two as an intermediate dataset with 10k samples. The results are shown above. The performance of the mixture goes in between the two datasets, with consistent trend as in the original finding (G-Vendi captures relative diversity in reasoning patterns, while embedding operates in an opposite direction).
>
> In addition, we note that the experiments in Table 2 are designed to analyze the feature-level difference between gradient and embedding diversity, by following the two extreme data generation pipelines for didactic purposes. The sensitivity of G-Vendi against baseline metrics is best shown in our main correlation analyses in Section 2.3.1 and Section B, where we consider a realistic data pool and sample many subsets in a continuous spectrum of diversity.
>
> ## Larger proxy models?
> For the practical applicability of G-Vendi, we assess that using a small proxy model (of ≤ 1B scale) is a fundamental necessity — as larger models would lead to proportionately more resources, just to collect the gradients. In this sense, the fact that G-Vendi can highly correlate with performance even while using a small proxy model is a strong appeal for our approach!
>
> Additionally, for other aspects of our experiments such as the main results for Prismatic Synthesis (Section 3.2) or ablation study on student models (Section 2.3.4), we leverage diverse set of models both in terms of model family (Llama and Qwen) and scale (up to 8B).
>
> Again, thank you for the positive and thoughtful feedback!

---

> > ### Comment · Reviewer_94F8 · 2025-08-06
> >
> > Thank the authors for their detailed response. The rebuttal addresses my concerns. I will hold my original rating.

---

> > > ### Author Response · Authors · 2025-08-06
> > >
> > > Thank you again for the helpful comments and positive feedback!

---

### Official Review · Reviewer_zTrb · 2025-07-07

**Clarity:** 3
**Significance:** 3
**Originality:** 3
**Rating:** 5
**Confidence:** 3

**Summary:**

The paper investigates the role of data diversity in improving generalization for LLMs, especially in reasoning tasks. The authors introduce a new metric, G-Vendi, which quantifies data diversity based on the entropy of model-induced gradients. This measure is shown to be strongly correlated with OOD generalization performance even when using a small off-the-shelf proxy model. Building on this insight, the paper proposes Prismatic Synthesis, a gradient-space-driven synthetic data generation framework. This framework identifies underrepresented gradient clusters and selectively samples new data points that enrich the gradient space. Using this technique, the authors construct PrismMath and PrismNLI, synthetic datasets that outperform several state-of-the-art models on a range of challenging benchmarks.

**Questions:**

see Weaknesses part

**Ethical Concerns:**

["NO or VERY MINOR ethics concerns only"]

**Final Justification:**

During the rebuttal, the authors provided detailed clarification regarding task similarity, overhead, scalability, and hyperparameter selection. I increased the score to accept.

**Limitations:**

yes

**Quality:**

3

**Strengths And Weaknesses:**

## Strengths
- Understanding what makes training data effective for reasoning is a fundamental question in LLM development
- The emphasis on gradient-based representations offers a new lens for understanding data quality. Specifically, G-Vendi metric is theoretically motivated and practically effective.  Prismatic Synthesis introduces a simple yet effective loop to diversify data based on underrepresented gradient space regions.
- PrismMath-7B outperforms models trained on data from larger generators, demonstrating real utility


## Weakness

- Iteration specification: It is not specified how many iterations are used in Prismatic Synthesis (line 248)

- There may be fundamental limits to how well gradient entropy on training data can predict reasoning ability on unseen benchmarks, particularly when training and evaluation domains differ significantly. The strong correlation observed may be partly due to domain similarity. For instance, the math experiments use OpenR1-Math as the seed dataset and evaluate on similar math reasoning tasks. Can G-Vendi/Prismatic Synthesis's effectiveness extend to other domains like code reasoning, logical reasoning, etc?  The connection between gradient entropy and downstream task performance is not very clear.

- There is limited discussion on failure modes or cases where G-Vendi might underperform

- Prismatic Synthesis overhead: The iterative clustering and generation process likely incurs high computational costs compared to simpler data selection methods, but the paper provides no direct comparison of computational overhead between Prismatic Synthesis and baseline approaches.

- G-Vendi scalability: Computing gradient entropy for large datasets (e.g., the 100k samples in Figure 2) may be computationally expensive. While the authors mention O(d²|D|) complexity, practical wall-clock time and memory requirements are not provided.


- Dimensionality reduction: The choice of 1024 dimensions for random projection (line 93) appears arbitrary without justification.
- Clustering parameters: Important hyperparameters like k=1% of data pool size and retaining the top 50% sparsest clusters lack justification

- Scale constraints: Experiments use relatively small models (1B-7B parameters); behavior at larger scales is uncertain

---

> ### Author Rebuttal · Authors · 2025-07-30
>
> We thank the reviewer for the positive assessment of our work and the thought feedback! We are glad you found our work to tackle fundamental problem in LLM development, our G-Vendi to be theoretically well grounded and effective, and our work to be of real utility. Below, we address each of your concerns.
>
> ## How many iterations were used?
> We do not fix the number of iterations, but limit the number of new samples generated at each iteration to be 10k (that is, after generating 10k samples, we move on to the subsequent step 3 in line 245). We iterate this process until we reach the target size of 1M. We will add this to the implementation details section in the revised version.
>
> ## On the notion of task similarity
> Thank you for pointing this out. You are correct that our setup assumes relevance between the training and evaluation tasks — e.g., training on math problems to improve model’s math reasoning skills — while not assuming any access to the evaluated benchmarks. This level of loose relevance is both realistic and practical, particularly in the context of curating reasoning data for LLMs — as even the state-of-the-art reasoning datasets  (e.g. Open-R1, MetaMath, SynLogic) specializes to specific domains in improving reasoning capabilities.
>
> But then, is reasoning data all about aggregating more datapoints for a specific domain? **Our experiments suggest the opposite — even when focusing on a specific task, the diversity of reasoning patterns significantly impacts model performance. While it is true that we use Open-R1 as seed with the goal of improving math reasoning, we still see a significant improvement from not only the Open-R1 itself, but also larger datasets with manual curation, by strategically improving diversity.** Overall, these results show that domain relevance, while generally important, is only an auxiliary factor behind our strong results.
>
> In terms of task breadth, we apply Prismatic Synthesis to two widely different tasks, namely generative math reasoning tasks based on long CoT, and discriminative NLI task with limited label space. We are already working on further application of Prismatic Synthesis beyond these two domains, which we believe is beyond our current scope that primarily introduce data diversity measure in a principled framework.
>
> ## Potential failure modes of G-Vendi
> Extending our discussion in Appendix D, we note that G-Vendi is not a "universal” metric that can predict whether one model would perform better on a specific benchmark. Importantly, our measure of model performance (with which G-Vendi strongly correlates) is an aggregation of metrics across multiple benchmarks, rather than targeting a specific test set. If one’s goal is to optimize to a specific benchmark, minimizing the distance between training distribution and the benchmark distribution could be more effective than increasing gradient diversity. But we view this as a difference by design, rather than an inherent limitation of G-Vendi.
>
> We also note that the gradient representation can often be hard to interpret (unlike examples shown in the appendix). We will include hard-to-interpret examples in our revision as well.
>
> ## Prismatic Synthesis Overhead
> We iterate gradient collection and clustering on every 10k generations. Since the projected gradient prior to the current steps are already saved in a storage, we only need to collect the gradients for the new 10K samples using a 0.5B proxy model. For clustering, we specifically implement Lloyd’s algorithm on GPU; since prior samples are already almost clustered, adding new 10K samples to the pool typically does not require more than 3 E-M steps in the Lloyd’s algorithm. Overall, we find that gradient collection on 1 H100 node typically takes less than 1 minute per step, and the clustering takes no more than 1.5 minutes per step in our experiments. Therefore, the computational overhead in Prismatic Synthesis is negligible compared to the the cost of distilling from larger models (which could take more than 1 hour on 1 H100 node for 10K samples with 32K max sequence length), which all baseline curation approaches share.
>
> In addition, we show in Table 4 that Prismatic Synthesis is substantially more model-efficient — our dataset outperforms state-of-the-art datasets generated via R1-671B, which is 21 times larger than our data generator (R1-32B). The efficiency gain from using a order-of-magnitude smaller teacher far outweighs that of the computational overhead describe above, adding to the practicality of our approach.
>
> ## G-Vendi Scalability
>
> | Setup                     | Wall-clock time | Max VRAM usgae |
> |---------------------------|:---------------:|:--------------:|
> | 100k samples, 1 H100 Node |     ~ 8m 30s    |     ~ 20GB     |
>
> As suggested by the reviewer, we measure the computational requirements to compute G-Vendi score. For the dataset size of 100k on 1 H100 node, G-Vendi computation took approximately 8m 30s. Most of the runtime is spent on iterating over the dataset and encoding each sample, which even the embedding-based baselines need to go through as well. Notably, aggregating Vendi score out of the collected gradients took less than 20 seconds, despite its theoretical complexity of O(d^2|D|). For memory requirements, the computation took no more than 20GB of GPU ram. This contrasts with using a 7B embedding model (as done in our baseline setup), which took 14GB of ram just to load the model on the GPU — indicating the efficiency of using small proxy model in G-Vendi. We appreciate the suggestion and will include this in the revision.
>
> ## Scale Constraints
> We would like to note that the nature of our analyses requires hundreds of iterations of experiments under a controlled setup, which can realistically be implemented with reasonable size models. **Importantly, as we primarily investigate diversity of training data, we study the effect of data scale with respect to diversity, incorporating experiments from small controlled setup (e.g. 10k) to a large scaling long CoT training (e.g. 1M).** But we also agree that understanding the impact of data diversity additionally with relation to model scale (e.g. > 70B) is an interesting and less-explored agenda, which we hope our work sets a foundation for.
>
> ## How was the projection dimension chosen?
> The dimension was empirically chosen based on our preliminary study on the correlation between G-Vendi scores with different projection dimensions on NLI data pool.
>
> | $d$  | 256   | 1024  | 8192  |
> |------|-------|-------|-------|
> | 256  | 1     | 0.940 | 0.922 |
> | 1024 | 0.940 | 1     | 0.974 |
> | 8192 | 0.922 | 0.974 | 1     |
>
> The results on 10k subsets are shown above. We find that dimensions beyond 1024 are very highly correlated — that is, using 1024-dim projection leads to 8x reduction in storage requirement than 8192, while almost perfectly preserving the relative positions of subsets. But we also find that further decreasing the projection size could harm the correlation against larger dimensions. Based on these findings we set projection dimension to 1024 in the subsequent experiments.
>
> ## Clustering Parameters
>
> | k               | 1%             | 10%            | 20%            |
> |-----------------|----------------|----------------|----------------|
> | Yield / G-Vendi | 0.288 / 189.34 | 0.275 / 185.15 | 0.294 / 178.15 |
>
> | N                   | k / 4          | k / 2          | k * 3/ 4       |
> |---------------------|----------------|----------------|----------------|
> | Yield / G-Vendi     | 0.152 / 190.32 | 0.288 / 189.34 | 0.463 / 168.42 |
>
> Thank you for the suggestion. We conduct 2 ablation studies on the impact of values of $k$, the number of clusters (defined as ratio of number of all samples) and $N$, the number of sparsest clusters to leave in Prismatic Synthesis. Using our math data generation pipeline, we generate 50k datasets with different values of k and N, and report the yield (ratio of samples passing the cluster filtering stage) and the G-Vendi score. When ablating $k$, we fix $N = k / 2$ as in our default setup.
>
> The results are shown in the above tables. We first find that our pipeline is robust to the specific values of $k$, exhibiting no conspicuous trend in G-Vendi or Yield. In addition, the result shows that the value of $k = 1\%$ leads to small improvement in the G-Vendi compared to larger values. On the impact of N, we see a tradeoff between yield / G-Vendi -- lower N leads to lower yield (i.e. needs more samples to get to a fixed data size), but achieves higher diversity on a fixed data size. These two effects ultimately offset each other in a long run (thus as reasonable values of $N$ would suffice), but we find that taking the middle value of $k/2$ adds to the stability by balancing the yield and diversity improvement.
>
>
> We thank the reviewer again for the positive and constructive suggestions!

---

> > ### Comment · Reviewer_zTrb · 2025-08-04
> >
> > Thanks for the detailed analysis about task similarity, overhead, scalability, and hyperparameter selection. My concerns have been addressed, and I will increase the score.

---

> > > ### Author Response · Authors · 2025-08-04
> > >
> > > Thank you again for the positive and thoughtful review! Your suggestions were of great help improving our work.

---

### Decision · Program_Chairs · 2025-09-17

**Decision:**

Accept (spotlight)

**Comment:**

This paper investigates how the diversity of synthetic training data impact generalization in LLM reasoning. It introduces a gradient based metric called G-Vendi and shows that data diversification with this metric could improve generalization. This is an important and useful topic for LLM training and reasoning. The proposed method is well motivated and the improvements clearly demonstrated by the experiments with comparison to alternative methods. The gradient based similarity and diversification in the gradient spaces provides interesting alternative angles.